# DictPFL: Efficient and Private Federated Learning on Encrypted Gradients

## Abstract

Federated learning (FL) enables institutions to collaboratively train machine learning models by aggregating local gradients without sharing sensitive data. However, sharing gradients still poses privacy risks, e.g., gradient inversion attacks. Homomorphic encryption (HE) is commonly used in FL to encrypt gradients at the data owner's side, enabling secure aggregation without decryption on the server. Existing HE-based FL methods are either fully encrypted or selectively encrypted: the former ensures privacy but incurs high overhead, while the latter improves efficiency by partially encrypting gradients, leaving shared unencrypted gradients vulnerable. To enable efficient and private FL, we propose DictPFL, a framework that encrypts shared gradients while keeping most gradients local without the need for sharing all, while preserving the performance of global gradient aggregation. DictPFL comprises two modules: Decompose-for-Partial-Encrypt (DePE) and Prune-for-Minimum-Encrypt (PrME). In DePE, we decompose pre-trained model weights into a dictionary and a lookup table. Only the gradients of the lookup table are encrypted and aggregated securely while the dictionary remains fixed and is not transmitted for aggregation. In PrME, we aim to further minimize the encrypted parameters with an encryption-aware pruning technique that ensures a consistent pruning mask across clients by leveraging the history of global gradients. Experimental results demonstrate that DictPFL significantly reduces communication overhead by 402 to 748 times and speeds training by 28 to 65 times compared to fully encrypted method. It also outperforms state-of-the-art selectively encrypted gradient by lowering overhead by 51 to 155 times and accelerating training by 4 to 19 times.

## 1. Introduction

Federated Learning (FL) (Shokri & Shmatikov, 2015) was introduced to enable collaborative training of a shared machine learning model among different data owners (e.g., hospitals or banks), where model gradients (or weights), rather than raw data, are shared to address privacy concerns. However, even sharing gradients poses privacy risks, as attackers could potentially exploit this information. For instance, model inversion (or gradient inversion) attacks (Zhu et al., 2019; Shi et al., 2023) have demonstrated the feasibility of reconstructing a client's original training data from the gradients shared by clients. In such scenarios, the server or users with access to the server can act as potential attackers.

To protect the privacy of clients' gradients during aggregation and enable private FL, various privacy-preserving primitives such as Differential Privacy (DP) (Truex et al., 2019; 2020; Sun et al.), Secure Multiparty Computation (MPC) (Bonawitz et al., 2017; So et al., 2022), and Homomorphic Encryption (HE) (Zhang et al., 2020; Fang & Qian, 2021; Jiang et al., 2021; Jin et al., 2023) have been utilized. Among these methods, HE is especially appealing in cross-silo settings (Zhang et al., 2020; Fang & Qian, 2021; Jiang et al., 2021; Jin et al., 2023), as it provides non-interactive privacy protection without the accuracy-privacy tradeoff associated with DP and without requiring the assumption of non-colluding servers, as in MPC. In HE-based privacy-preserving federated learning, locally updated gradients are encrypted by clients before sharing with the server, allowing the server to perform homomorphic aggregation directly on ciphertexts. Despite its security benefits, HE introduces significant overhead: ciphertext expansion increases communication costs by 1 to 3 orders of magnitude, while encryption, decryption, and homomorphic aggregation impose high computational costs (Zhang et al., 2020; Jin et al., 2023).

To mitigate the above issues of HE-based FL, one direction is to reduce the number of encrypted gradients, as fewer ciphertexts result in lower HE-related communication and computation overheads. The state-of-the-art literature, FedML-HE (Jin et al., 2023) as shown in Figure 1 (a), implements a *Select-and-Encrypt (SaE)* strategy: clients pre-calculate sensitivity scores for parameters, encrypting only the gradients of top 10% sensitive parameter while transmitting the remaining 90% less-sensitive parameters in plaintext. However, unencrypted parameters still suffer from risk exposure, e.g., Zhu et al. (2019) show that accessing 30% of gradients enables training data reconstruction, leading to privacy issues. Furthermore, its pre-calculated sensitiv-

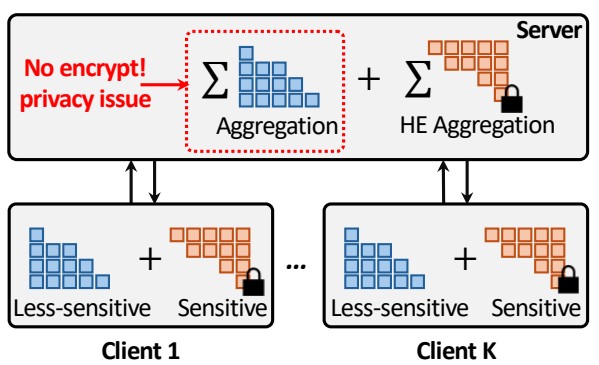

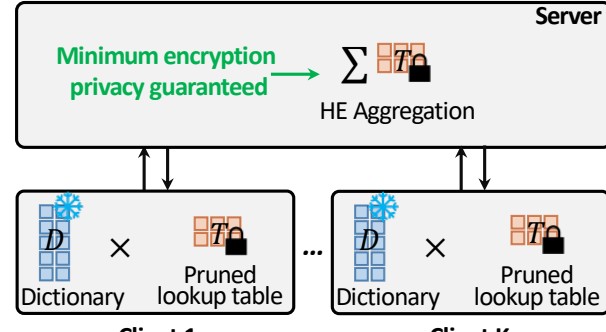

(a) Prior HE-based PPFL with *Select-and-Encrypt (SaE)* strategy

(b) Our DictPFL with *Decompose-for-Partial-Encrypt (DePE)* and *Prune-for-Minimum-Encrypt (PrME)* strategy

*Figure 1.* (a) Prior HE-based PPFL (Jin et al., 2023) encrypts only sensitive gradients. The less-sensitive weights are shared without encryption, which may lead to privacy concerns. (b) In contrast, our DictPFL approach minimizes encryption while ensuring privacy guarantees through the Decompose-for-Partial-Encrypt (DePE) and Prune-for-Minimum-Encrypt (PrME) strategies. DePE involves decomposing gradients into a frozen dictionary and a trainable lookup table, with only the encrypted lookup table being shared for aggregation. PrME further prunes the lookup table parameters on the client side to reduce encryption costs.

ity scores often are limited to capture dynamic sensitivity during training, as parameter updates alter their privacy-sensitivity. Thus, encrypting all gradients sent to the server remains essential to prevent leakage.

The *Select-and-Encrypt (SaE)* strategy inevitably exposes privacy risks due to shared unencrypted data, although achieving fewer communication and faster training over the previous fully encrypted methods. To address this challenge, we propose DictPFL as shown in Figure 1 (b), which ensures that the shared parameters are fully encrypted to guarantee privacy while mimizing the shared parameters by two modules: *Decompose-for-Partial-Encrypt (DePE)* and *Prune-for-Minimum-Encrypt (PrME)*. DePE decomposes the pre-trained model into a globally consistent dictionary, which is identical across all clients, and a lookup table, where each client trains independently. Only the encrypted gradients of the lookup table are transmitted to the server for aggregation, while the globally consistent dictionary remains frozen and is never transmitted. *PrME* is further proposed to minimize the encrypted lookup tables. Unlike plaintext-level pruning techniques in FL (Aji & Heafield, 2017; Li et al., 2021; Bibikar et al., 2022), where clients often perform local-specific pruning and share pruned indices for aligned aggregation on the server side, or where the server directly prunes gradients, HE-based FL presents new challenges for gradient pruning: encrypted gradients are difficult to align and prune during aggregation. Our proposed *PrME* addresses this issue and ensures consistent pruning across all clients without requiring encrypted pruning. This is achieved by leveraging gradient history: instead of relying on local gradient magnitudes, all clients prune their gradients based on a shared reference, thereby aligning pruning indices across clients. Additionally, dynamic probabilities are assigned to the pruned parameters, allowing for their

potential reintroduction in future rounds and mitigating the negative effects of pruning. Since the pruned lookup tables are significantly smaller than the full model weights, and all transmissions are encrypted, this approach substantially reduces the number of ciphertexts without compromising privacy.

Extensive experiments demonstrate that DictPFL achieves substantial performance improvements over the state-of-the-art FedML-HE (Jin et al., 2023) across various tasks, including (i) image recognition, (ii) text classification, and (iii) text generation. Specifically, compared to private fully encrypted frameworks (Roth et al., 2022), DictPFL reduces communication overhead by 402 to 748 times and accelerates training by 28 to 65 times. It also outperforms the selectively encryted method FedML-HE, by lowering overhead by 51 to 155 times and speeding up training by 4 to 19 times.

## 2. Background and Motivation

### 2.1. Privacy-preserving Federated Learning

Federated learning enables collaborative training among distributed clients without directly sharing datasets. In this framework, the clients train their models locally and send the gradients (or model updates) to a central server, which aggregates these gradients using algorithms like FedAvg (McMahan et al., 2017) and FedSGD (Shokri & Shmatikov, 2015). However, the direct exposure of local gradients to the server poses severe privacy risks (Mothukuri et al., 2021). For instance, with access to client's local gradients, the server can perform model inversion attacks (Zhu et al., 2019; Hitaj et al., 2017; Shi et al., 2023) to reconstruct the client's dataset.

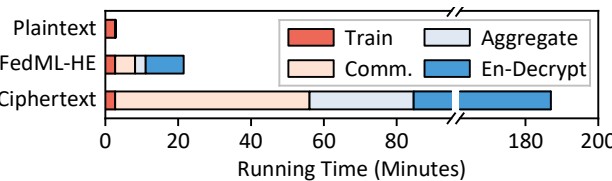

*Figure 2.* Training time breakdown in plaintext, ciphertext and FedML-HE (Jin et al., 2023) settings for a ViT model on GTSRB.

Several methods have been proposed to protect the gradients transmitted between clients and the server. One strategy employs Differential Privacy (DP)(Truex et al., 2020; 2019; Sun et al.) by injecting noise into the gradients before sharing them. Although DP imposes minimal computational overhead, it inevitably degrades model performance because of the added noise. Secure Multi-Party Computation (MPC) (Fereidooni et al., 2021), requires $N \geq 2$ non-colluding servers to jointly aggregate client gradients in a privacy-preserving way, where each server can only access encrypted gradient shares, not the original values. However, this reliance on multiple non-colluding servers makes it unsuitable for single-server settings.

Another approach leverages Homomorphic Encryption (HE) to encrypt gradients on the client side, enabling the server to aggregate encrypted gradients without decryption. Prior HE-based FL methods either use limited schemes like additive homomorphic encryption (AHE) (Zhang et al., 2020; Fang & Qian, 2021; Jiang et al., 2021), which lack support for general aggregation, or employ computationally impractical fully homomorphic encryption (FHE) (Brakerski et al., 2014; Cheon et al., 2017; Chillotti et al., 2020). While platforms like IBM FL (IBM, 2022) and Nvidia Flare (Roth et al., 2022) have explored the integration of FHE, they fail to address its significant overheads. As shown in Figure 2, HE operations dominate training time, and ciphertext size substantially increases communication costs.

### 2.2. Efficient HE-based Federated Learning

Recently, many efforts have been made to improve the efficiency of HE-based FL. These optimization strategies can be broadly classified into two categories, i.e., encryption scheme optimization and algorithmic optimization.

Quantization (Zhang et al., 2020; Xu et al., 2021; Han & Yan, 2023) and Packing (Zhang et al., 2020; Aono et al., 2017; Liu et al., 2019) are widely studied techniques within the realm of encryption scheme optimization for HE-based FL. Quantization reduces communication costs by converting high-precision gradients into low-precision values. On the other hand, packing, also referred to as batching, focuses on consolidating multiple local gradients into a single plaintext, significantly reducing the number of plaintexts that need to be encrypted and sent.

Algorithmic optimization involves tailoring efficient strategies based on the characteristics of the machine learning model, and our DictPFL falls into this category. The state-of-the-art work, FedML-HE (Jin et al., 2023) proposes to selectively encrypt the gradients based on privacy-sensitive scores, i.e., *Select-and-Encrypt (SaE)*, as shown in Figure 1 (a). However, it suffers from several critical limitations. First, privacy-sensitive scores are computed once before training and remain static throughout the training process. This static approach fails to account for how weight sensitivity changes during training, because weights classified as non-sensitive on the initialized model may later become critical for privacy protection. Most critically, it cannot ensure complete privacy protection. Since only the gradients of selected parameters are encrypted, the remaining gradients are transmitted in plaintext, leading to inevitable information leakage and making it impossible to guarantee privacy protection regardless of which gradients are selected for encryption. Additionally, as illustrated in Figure 2, although FedML-HE substantially reduces the communication overhead and HE operations (including aggregation, encryption, and decryption) by a factor of ten when only the top 10% of sensitive parameters are encrypted, these overheads induced by ciphertexts are still primary bottlenecks in the training process.

### 2.3. Motivation

As illustrated in Figure 2, communication and computation overheads caused by ciphertexts becomes the main bottleneck in HE-based federated learning. Although state-of-the-art FedML-HE (Jin et al., 2023) attempts to improve efficiency by selectively omitting encryption for partial parameters, it not only compromises privacy but also continues to struggle with significant HE-induced communication and computation overheads. To achieve higher efficiency without sacrificing privacy, we focus on reducing the total number of trainable parameters. Guided by this principle, we propose DictPFL, which employs two strategies: *Decompose-for-Partial-Encrypt (DePE)* (Section 4.1) to decompose gradients and *Prune-for-Minimum-Encrypt (PrME)* (Section 4.2) to prune the gradients of parameters that exhibit minimal updates.

## 3. Preliminaries

### 3.1. System Overview: Federated Learning with HE

Same with FedML-HE (Jin et al., 2023), the workflow of HE-based privacy-preserving federated learning begins with clients utilizing a trusted key authority to generate a public-secret HE key pair. During each training iteration: (1) clients compute local gradients; (2) these gradients are encrypted with the public key and transmitted to the server; (3) the server aggregates the encrypted gradients; (4) the aggregated

ciphertext is broadcast back to clients, who decrypt it using their secret keys and update their local models with the decrypted result.

### 3.2. Threat Model

We consider a semi-honest adversary $\mathcal{A}$ that may corrupt the server, which is the same as the setting of FedML-HE (Jin et al., 2023). While $\mathcal{A}$ follows the protocol, it attempts to infer private information from benign participants. Security guarantees ensure $\mathcal{A}$ learns no information from the data of clients.

## 4. DictPFL

### 4.1. Decompose-for-Partial-Encrypt (DePE)

**Overview.** Model weight decomposition, representing a weight matrix $W$ as a linear combination of vectors from a compact dictionary $D$ and a sparse lookup table $T$, is a proven strategy for parameter reduction in inference (Lou et al., 2023; Bagherinezhad et al., 2017). The key insight lies in reducing the inherent redundancy in weight parameters: correlated parameters can be represented as sparse linear combinations of a dictionary of vectors. We adapt this principle to HE-based federated learning, where reducing the dimensionality of trainable parameters, directly minimizes the number of ciphertexts.

**Constructing $W$ with $D$ and $T$.** Figure 3 demonstrates the construction of the weight matrix $W \in \mathbb{R}^{n \times m}$ using a dictionary $D \in \mathbb{R}^{n \times r}$ and lookup table $T \in \mathbb{R}^{r \times m}$. Each column vector $W[:][i]$ of $W$ is derived through a linear combination of the $r$ vectors in $D$, weighted by the corresponding scalars in the $i$-th column of $T$, denoted $T[:][i]$. This process is formally expressed by:

$$W[:][i] = \sum_{k=0}^{r} D[:][k] \cdot T[k][i] \tag{1}$$

By reducing $r$, the dictionary size, we effectively decrease the number of trainable parameters, thereby reducing the communication overhead associated with ciphertexts.

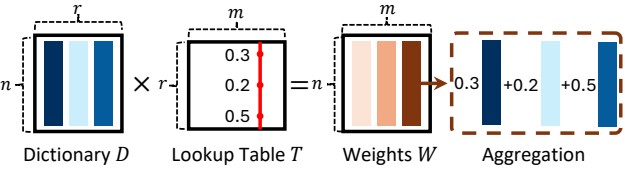

Dictionary $D$    Lookup Table $T$    Weights $W$    Aggregation

*Figure 3.* Representing the weight matrix $W$ with dictionary $D$ and lookup table $T$. For instance, given $r = 3$, the $i$-th column of $T$ is $[0.3, 0.2, 0.5]$, the $i$-th column of weights $W$ is represented by $W[:][i] = 0.3 \cdot D[:][0] + 0.2 \cdot D[:][1] + 0.5 \cdot D[:][2]$.

**Facorization of Dictionary and Lookup Tables.** To ensure that the dictionary $D$ contains critical and generaliz-

able weight vectors and remains constant across all clients, DePE leverages the knowledge encapsulated in pre-trained weights $W_0$. We employ a truncated SVD factorization to decompose $W_0$, which has dimensions $n \times m$, into a smaller dictionary $D$ and a lookup table $T'$. Specifically, $W_0$ is approximated as $U_r \Sigma_r V_r^\top$, where $U_r$, $\Sigma_r$, and $V_r^\top$ correspond to the top-$r$ singular values and vectors, thus reducing the dimensionality to $n \times r$ for $D$ and $r \times m$ for $T'$,

$$W_0 \approx U_r \Sigma_r V_r^\top \tag{2}$$

$$D, T' = SVD(W_0, r) = U_r \Sigma_r, V_r^\top \tag{3}$$

DePE initializes $D$ as $U_r \Sigma_r$ and $T'$ as $V_r^\top$ according to Equation 3. However, directly freezing $D$ and training $T'$ can lead to suboptimal performance due to the information loss inherent in SVD truncation, particularly when $r$ is much smaller than $m$ or $n$. To counteract this, we retain the pre-trained weight $W_0$ and initialize $T$ by zeroing out $T'$. This strategy allows for the construction of $W$ as $W_0 + D \cdot T$, with $D$ remaining static and shared among all clients, while $T$ is updated locally and aggregated on the server. By selecting a smaller $r$, we significantly reduce the communication overhead for encrypted parameters, as encryption is only required for the $r \times m$ entries in $T$.

### 4.2. Prune-for-Minimum-Encrypt (PrME)

As DePE training progresses, there is a decline in the number of parameters with large gradients, as demonstrated in Figure 4 (a). By the 50th training round, only a small subset of parameters still exhibit gradients exceeding $10^{-5}$, as shown in Figure 4 (b). Encrypting and transmitting all gradients to the server for aggregation, including those of parameters that no longer change significantly, introduces unnecessary redundancy. By enabling clients to selectively upload the substantial gradients, communication overhead can be dramatically reduced.

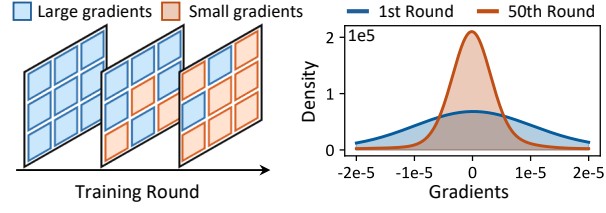

(a) Evolution of Gradients Magnitude    (b) Change in Gradients Distribution

*Figure 4.* (a) As training progresses, parameters that initially have large gradients may gradually transition to having smaller gradients. (b) Concurrently, the number of parameters with substantial gradients decreases significantly.

Existing gradients pruning methods in plaintext federated learning involve clients independently pruning their smallest local gradients before transmission to the server for

aggregate. Since clients possess different local gradients, they may prune parameters at different positions, necessitating the sharing of pruning indices with the server to ensure proper aggregation. However, implementing such methods to HE-based federated learning presents two fundamental challenges. First, encrypted indices force the server to perform non-linear operations (e.g., comparing encrypted indices to match) alongside linear operations (e.g., aggregation), a hybrid workflow that incurs prohibitive computational overhead (Zhang et al., 2024b). Second, the SIMD batching mechanism, which packs multiple plaintext gradients into several slots of a single ciphertext, renders index-specific operations infeasible. Since HE aggregation occurs slot-wise, gradients occupying the same slot across clients are combined automatically, regardless of their indices.

Figure 5 illustrates the challenges of pruned HE aggregation. Consider a scenario where client A encrypts and uploads gradients from positions 1 and 3, while client B encrypts and uploads gradients from positions 1 and 4. The server cannot perform correct aggregation because the ciphertext slots are misaligned, and the encryption prevents any coordination or realignment of the gradients. To ensure consistent gradient pruning accross clients, they require an identical metric for determining which gradients to prune. The optimal approach would involve clients pruning their local gradients based on current round global gradients. However, clients cannot access the current round global gradients until after sharing their complete local gradients with the server for aggregation. This creates a dilemma: clients cannot prune independently as it leads to inconsistencies, nor can they rely on global gradients to coordinate pruning.

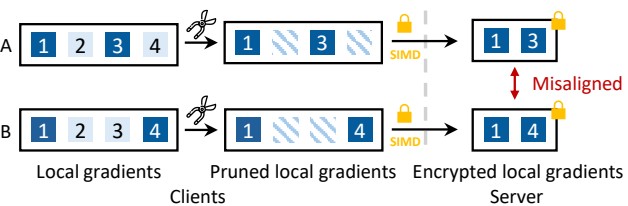

A

B

Local gradients    Pruned local gradients    Encrypted local gradients

Clients              Server

*Figure 5.* An example of failed aggregation due to different locations pruned by client A and client B.

**Temporal Inactivity Pruning (TIP).** To resolve this dilemma, clients require a shared pruning metric independent of the current round's global gradients. A straightforward solution is to base pruning decisions on the last round's global gradients, which are identical across clients and accessible before aggregation. Specifically, clients prune local gradients corresponding to parameters with the smallest $s\%$ magnitudes from the prior global gradients. However, parameters showing minimal activity in one round may experience significant updates in subsequent rounds, leading to unintended removal if pruning decisions rely exclusively on last round gradients. For instance, as illustrated in Fig-

ure 6 (b), the parameter with a small gradient magnitude in an earlier round may be pruned, despite its gradient resurgence in later rounds, as indicated in Figure 6 (a).

To mitigate the influence of transient fluctuations and retain critical gradients, we introduce a temporal windowing strategy that leverages the information from the previous $\tau$ consecutive rounds. Clients identify parameters whose gradients fall within the smallest $s\%$ across all $\tau$ rounds (pruning patience). Formally, the pruning mask for parameter $w_i$ at round $t$ is defined as:

$$M_{i,t} = \begin{cases} 0 & \text{if } \sum_{k=1}^{\tau} \mathbf{1}\left(|\delta w_{i,t-k}| < \theta_{s,t-k}\right) = \tau \\ 1 & \text{otherwise} \end{cases} \quad (4)$$

Here, $M_{i,t} = 0$ indicates pruning the local gradient of $w_i$, while $M_{i,t} = 1$ retains its local gradient for aggregation. The $\delta w_{i,t-k}$ denotes the global gradient of parameter $w_i$ at round $t - k$, and $\mathbf{1}$ is the indicator function. The threshold $\theta_{s,t-k}$ dynamically adapts as the $(100\text{-}s)$-th percentile of $|\delta w_{i,t-k}|$. As shown in Figure 6 (c), the pruning is postponed to a later round when gradients exhibit more stable behavior, thereby preserving gradients that regain significance after initially being considered for pruning.

**Holistic Reactivation Correction (HRC).** Although TIP reduces communication overhead whlie preventing premature pruning by chance, it still has an inherent limitation: once a parameter is pruned, its local gradients no longer participate in aggregation. Consequently, its global gradient magnitudes remain zero in subsequent rounds, effectively excluding it permanently. This irreversible pruning can hinder training convergence, as parameters with substantial gradients in later rounds may no longer be updated. For example, in Figure 6 (a), the example parameter may have siginificant gradients magnitude even after the 100th round.

To mitigate the performance loss caused by irreversible pruning, we propose a dynamic reactivation scheme, Holistic Reactivation Correction (HRC). Instead of permanently excluding pruned parameters, HRC assigns each pruned parameter $w_i$ a reactivation probability $p_i$, which is dynamically adjusted based on its aggregated global gradients $\delta w_{i,t}$ after reactivation:

$$p_i[t+1] = \begin{cases} p_i[t] \times \beta & \text{if } |\delta w_{i,t}| < \theta_{s,t} \\ \min\left(p_i[t]/\beta, 1\right) & \text{otherwise} \end{cases} \quad (5)$$

Here, $\beta$ is a decay factor less than 1. When a pruned parameter is reactivated, the client uploads its *accumulated* local gradients since the pruning round for aggregation and gets the current round's global gradients $\delta w_{i,t}$. This approach preserves small gradients that, while individually minor, can meaningfully accumulate over time, rather than discarding

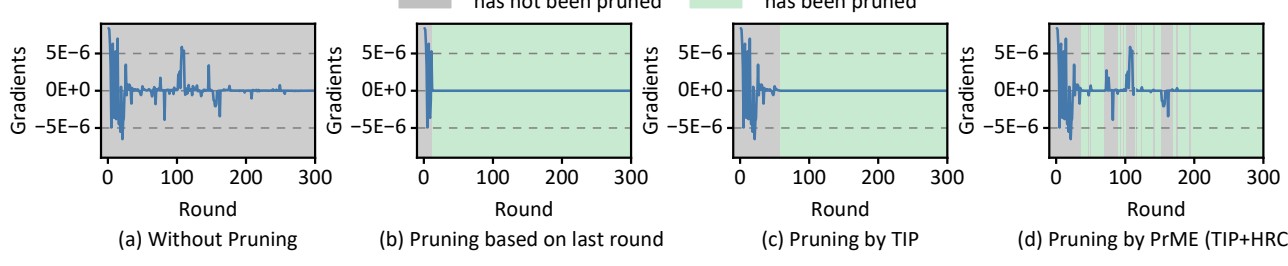

Figure 6. Evolution of a parameter's global gradients under different pruning strategies. Green background indicates the parameter is pruned (excluded from aggregation), while gray background indicates the opposite. Larger green areas reflect more overhead reduction. Closer alignment of gradient trends with the baseline (a) signifies preserved convergence performance.

these gradients, maintaining them locally for future aggregation helps convergence. If $|\delta w_{i,t}| < \theta_{s,t}$, indicating that the parameter's cumulative global gradients remain small even after reactivation, the reactivation probability $p_i$ decreases, discouraging further reactivation. Conversely, if $|\delta w_{i,t}| \geq \theta_{s,t}$, $p_i$ increases, encouraging the update of this parameter to rejoin aggregation. This adaptive mechanism mitigates information loss from premature pruning by flexibly adjusting the likelihood of reactivation. Although HRC introduces some uncertainty, consistency across clients can be easily maintained by preserving a shared random seed for the pruning mask.

## 5. Experimental Methodology

**Datasets.** We conduct experiments on three image classification tasks: CIFAR-10 (Krizhevsky et al.), GTSRB (Houben et al.), and Diabetic Retinopathy (Gulshan et al.), as well as AG's News (Zhang et al.) for sentence classification and MetaMathQA (Yu et al., 2023) for text generation. The experiments are performed under varying levels of data heterogeneity and with client numbers. We generate homogeneous data splits by randomly assigning training examples to individual clients without replacement. For heterogeneous settings, we simulate the data heterogeneity by sampling the label ratios from a Dirichlet distribution with a symmetric parameter, following the (Hsu et al., 2019). In both settings, each client holds the same number of samples, following (Kim et al., 2024).

**Models.** We perform DictPFL on multiple prevalent transformer-based models including, ViT (Dosovitskiy, 2020) designed for image recognition, BERT (Kenton & Toutanova, 2019), and TinyLlama (Zhang et al., 2024a) for natural language processing.

**Baselines.** We compare DictPFL with three baselines: FedHE-Full (Roth et al., 2022), which trains the whole model and encrypts all gradients; FedHE-Top2, fine-tuning only the last two layers; and FedHE-ML (Jin et al., 2023), which encrypts a subset of gradients (10% unless specified otherwise) while leaving the rest in plaintext.

**Evaluation Metrics.** We assess the efficacy of our proposed DictPFL by comparing its communication overhead, training time, and model accuracy against existing methods.

For privacy evaluation, we compare DictPFL with FedML-HE (Jin et al., 2023) in terms of potential privacy leaks. We utilize recovered image scores derived from $1 - \text{LPIPS}$, where the Learned Perceptual Image Patch Similarity (LPIPS) (Huang et al., 2021) measures discrepancies between reconstructed and original images. Therefore, higher scores indicate greater similarity and consequently, higher privacy risks.

**Hyperparameters.** Unless otherwise specified, we set the dictionary size $r$ to 4, the pruning ratio $s\%$ to 70%, the pruning patience $\tau$ to 3, and the reactivation probability scaler $\beta$ to 0.2. Detailed analysis of these hyperparameters are provided in Section 6.2.

**HE Implementation.** We adopt the CKKS homomorphic encryption scheme with bootstrapping (Cheon et al., 2017; 2019; 2018), implemented via OpenFHE (Badawi et al., 2022). We configure parameters for 128-bit security (see Appendix A.1) and leverage SIMD (Smart & Vercauteren, 2014) for parallelized ciphertext operations. Data encoding follows (Crockett, 2020). Experiments were conducted on an AMD Ryzen Threadripper PRO 3955WX processor (2.2GHz) with 125GB of memory.

## 6. Results

### 6.1. Main Results

**Comparison with Existing Works.** To demonstrate DictPFL's effectiveness, we compare it with other HE-based FL frameworks on the CIFAR-10, Diabetic Retinopathy, and GTSRB datasets using the ViT-16 model. Figure 7 illustrates a holistic comparison. Notably, DictPFL significantly and consistently reduces communication overheads compared to the baselines without sacrificing accuracy. Specifically, FedHE-Full has the highest communication demand. FedHE-Top2, which fine-tunes only the last two layers, shows reduced overhead but underperforms, because freezing most layers limits learning capacity, particularly on

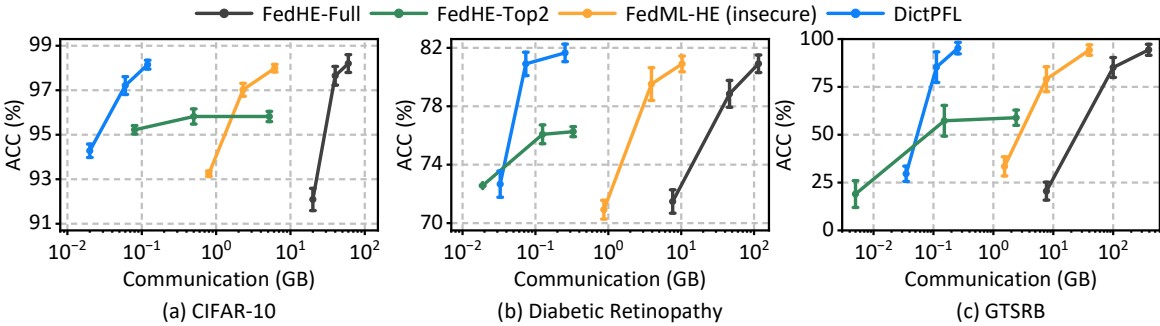

*Figure 7.* Efficiency comparison of different federated frameworks, in terms of accuracy versus communication overhead on three datasets using the ViT model. Higher efficiency is indicated by higher accuracy for the same communication or achieving the same accuracy with less communication, as shown by lines closer to the upper left corner. Communication is quantified by the total amount of data exchanged, including plaintexts and ciphertexts, during the training iterations.

datasets that diverge from those used in pre-training. For instance, it achieves only 58.9% accuracy on GTSRB versus DictPFL's 95.27%.

DictPFL achieves a 98.3% average reduction in communication overhead compared to the state-of-the-art FedML-HE (encrypt 10%), while maintaining the same level of accuracy. Although FedML-HE also reduces communication costs, it does so at the expense of privacy by exposing part of gradients in plaintext. DictPFL, on the other hand, fully preserves privacy. This is further demonstrated in Figure 8 (a), which highlights the vulnerability of FedML-HE to state-of-the-art gradient inversion attacks (Wen et al., 2022).

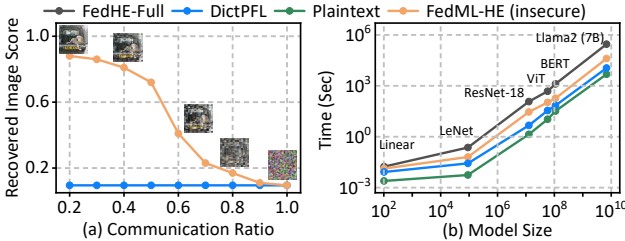

*Figure 8.* (a) Gradient inversion attack against FedML-HE and DictPFL. The communication ratio is the communication overhead relative to encrypting the full-size model gradients in FedHE-Full. (b) Comparison of communication overhead of DictPFL and baselines on models of different sizes.

In addition to ViT, we evaluate several other models, as shown in Figure 8 (b). The results show that DictPFL consistently outperforms the baselines across models of different scales. Compared with the fully encrypted baseline FedHE-Full, DictPFL reduces communication by 402 to 748 times and accelerates training by 28 to 65 times. It also outperforms the selectively encrypted baseline FedML-HE by lowering overhead by 51 to 155 times and speeding up training by 4 to 19 times.

**Breakdown Analysis.** In Figure 9, we break down the training time for various HE-based FL frameworks under both LAN and WAN settings. In FedHE-Full, where all gradi-

ents are encrypted, communication and ciphertext-related operations (encryption, decryption, and aggregation) dominate the training time. FedHE-Top2 reduces communication and ciphertext-related operations by fine-tuning the last two layers, but this comes at the cost of reduced accuracy, achieving only 58.9%. On the contrary, our proposed DePE and PrME techniques significantly reduce the number ciphertexts, resulting in a total training time that is 1 to 2 orders of magnitude lower than other baselines while maintaining a comparable level of accuracy.

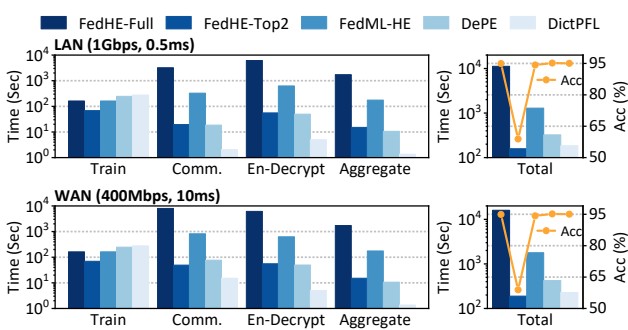

*Figure 9.* Training time breakdown of ViT on GTSRB.

### 6.2. Ablation Study

In this section, we explore the design space of DictPFL and study the impact of various settings on its performance. Unless otherwise specified, all experiments are conducted using the Diabetic Retinopathy dataset within a 3-client homogeneous setting within 10 rounds.

**Hyperparameters of DePE.** The dictionary size is a crucial hyperparameter in our DePE. A larger dictionary captures more comprehensive representations of gradients, enhancing accuracy but increasing overheads. As evidenced in Table 1, even a small dictionary with $r = 4$ achieves commendable training performance, e.g., an accuracy of 81.99%, close to the 82.74% achieved by FedHE-Full. This efficacy stems from the dictionary's ability to retain essential information

corresponding to the largest singular values.

*Table 1.* The results of DictPFL under different dictionary sizes $r$.

| $r$ | Accuracy (%) ↑ | Comm. (GB) ↓ | Time (min) ↓ |
|---|---|---|---|
| 2 | $74.26_{\pm 0.5}$ | 0.046 | $6.11_{\pm 0.1}$ |
| 4 | $81.99_{\pm 0.4}$ | 0.088 | $6.23_{\pm 0.1}$ |
| 8 | $82.67_{\pm 0.2}$ | 0.160 | $6.42_{\pm 0.2}$ |
| 16 | $82.71_{\pm 0.2}$ | 0.332 | $7.27_{\pm 0.1}$ |

**Hyperparameters of PrME.** In our Prune-for-Minimum-Encrypt (PrME), we explore the impact of varying the pruning ratio $s\%$ and pruning patience $\tau$. A higher $s\%$ results in more minor gradients being pruned, whereas a lower value preserves them. As shown in Figure 10, without PrME (prune 0%), training converges rapidly within 10 rounds, but each round incurs the highest communication cost. Pruning 70% drastically reduces communication overhead but significantly impacts accuracy. By contrast, pruning 20% preserves accuracy but results in far less communication reduction compared to the 70% pruning scenario. Notably, with our HRC reactivation scheme, immaturely pruned gradients in earlier rounds can be selectively reintroduced in later rounds. This enables the model to achieve accuracy similar to the 20% pruning scenario while achieving the communication efficiency of the 70% pruning ratio.

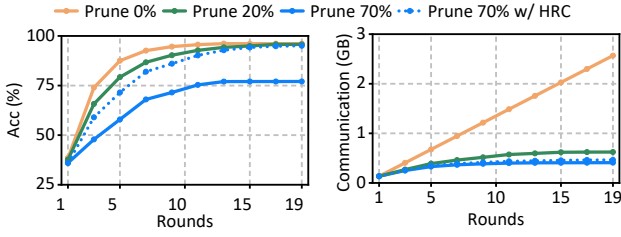

*Figure 10.* Ablation on pruning ratio under $\tau = 3$ and $\beta = 0.2$.

Table 2 studies different pruning patience $\tau$. Higher $\tau$ values delay the pruning of gradients, reducing accuracy degradation but limiting the communication reduction. Notably, setting $\tau = 3$ already results in small accuracy loss. This resilience can be attributed to our HRC, which mitigates the impact on accuracy by reintroducing pruned gradients, effectively correcting errors over time.

*Table 2.* Various pruning patiences under $s\% = 70\%$ and $\beta = 0.2$.

| $\tau$ | Accuracy (%) ↑ | Comm. (GB) ↓ | Time (min) ↓ |
|---|---|---|---|
| 1 | $80.55_{\pm 0.6}$ | 0.001 | $6.26_{\pm 0.1}$ |
| 3 | $82.29_{\pm 0.3}$ | 0.003 | $6.36_{\pm 0.1}$ |
| 5 | $82.67_{\pm 0.2}$ | 0.160 | $6.42_{\pm 0.2}$ |
| 10 | $82.77_{\pm 0.3}$ | 0.474 | $6.92_{\pm 0.1}$ |

Table 5 in Appendix B.1 showcases that our PrME works well under different reactivation probability scaler $\beta$.

**Different Number of Clients.** We assess the performance of DictPFL in environments with varying numbers of clients. The findings, presented in Table 3, demonstrate that DictPFL performs effectively and consistently across settings with different client counts.

*Table 3.* The results of DictPFL under client numbers.

| Clients | Accuracy (%) ↑ | Comm. (GB) ↓ | Time (min) ↓ |
|---|---|---|---|
| 3 | $82.67_{\pm 0.2}$ | 0.160 | $6.42_{\pm 0.2}$ |
| 5 | $82.64_{\pm 0.1}$ | 0.092 | $3.70_{\pm 0.1}$ |
| 10 | $81.94_{\pm 0.4}$ | 0.046 | $1.85_{\pm 0.1}$ |
| 20 | $81.82_{\pm 0.4}$ | 0.041 | $0.93_{\pm 0.2}$ |

**Different Heterigious Level.** Unsurprisingly, DictPFL performs better in homogeneous settings than in heterogeneous settings. As the table 4 shows, we evaluated DictPFL in various heterogeneous settings under different Dirichlet distributions from 0.3 to 0.9 and compared it with a homogeneous setting. The results indicate that DictPFL's performance remains stable across different heterogeneous dataset splits. Specifically, a smaller $\alpha$ (more heterogeneous) requires more communication size and training time to achieve comparable accuracy to a larger $\alpha$ (less heterogeneous).

*Table 4.* The results under different heterogeneous settings.

| $\alpha$ | Accuracy (%) ↑ | Comm. (GB) ↓ | Time (min) ↓ |
|---|---|---|---|
| 0.3 | $79.62_{\pm 0.4}$ | 0.103 | $6.22_{\pm 0.2}$ |
| 0.6 | $80.28_{\pm 0.2}$ | 0.145 | $6.44_{\pm 0.1}$ |
| 0.9 | $82.06_{\pm 0.3}$ | 0.151 | $6.45_{\pm 0.2}$ |
| $\infty$ | $82.67_{\pm 0.2}$ | 0.160 | $6.42_{\pm 0.2}$ |

### 6.3. Other Experiments

The results for text tasks, including classification and generation tasks are in Appendix B.2. DictPFL outperforms all the baselines on language tasks.

## 7. Conclusion

In this work, we present DictPFL, a novel framework for efficient HE-based FL. By decomposing model weights into a static dictionary and a trainable lookup table through Decompose-for-Partial-Encrypt (DePE), and further optimizing with Prune-for-Minimum-Encrypt (PrME), DictPFL significantly reduces encrypted gradient transmission without compromising privacy. Compared to the fully encrypted method, DictPFL lowers communication overhead by 402 to 748 times and speeds training by 28 to 65 times. It also outperforms selectively encrypted FedML-HE, reducing overhead by 51 to 155 times and accelerating training by 4 to 19 times, while preserving model performance and eliminating privacy risks from partial plaintext gradient transmission.

## Impact Statement

The paper introduces DictFPL, a method designed to reduce the computational and communication overheads associated with protecting federated learning shared weights using homomorphic encryption. This approach enhances privacy protections without compromising accuracy, making it a more feasible solution for large-scale, real-world applications. By ensuring that sensitive weights remains private, DictFPL can accelerate the adoption of federated learning across industries such as healthcare, finance, and beyond, while fostering trust in AI systems and promoting global data privacy.

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

## A. Experimental Setup

### A.1. HE Parameters

We configure the CKKS scheme with a cyclotomic ring dimension $N = 2^{16}$, ciphertext modulus of 1555 bits, and multiplicative depth $L = 12$ to ensure 128-bit security under the Homomorphic Encryption Standard (Albrecht et al., 2021). Each ciphertext contains $N/2 = 32{,}768$ slots for parallelized SIMD operations.

## B. More Experiments

### B.1. Different reactivation probability scale $\beta$

Table 5 studies different reactivation probability scalers $\beta$. The result showcase the our PrME works well under different $\beta$.

*Table 5.* Ablation study on $\beta$ under $s\% = 70\%$ and $\tau = 3$.

| $\beta$ | Accuracy (%) ↑ | Comm. (GB) ↓ | Time (min) ↓ |
|---|---|---|---|
| 0.2 | $82.29_{\pm 0.3}$ | 0.003 | $6.36_{\pm 0.1}$ |
| 0.5 | $82.37_{\pm 0.3}$ | 0.007 | $6.36_{\pm 0.1}$ |
| 0.8 | $82.55_{\pm 0.2}$ | 0.031 | $6.39_{\pm 0.2}$ |

### B.2. Performance on NLP tasks.

Table 6 shows that DictPFL significantly improves efficiency in both sentence classification and generation (instruction tuning) tasks. For the generation task, we train on the MetaMathQA (Yu et al., 2023) dataset and evaluate on GSM8K (Cobbe et al., 2021), focusing on mathematical reasoning. These gains are especially pronounced in larger models, where DictPFL reduces training time by 99.4% percent for TinyLlama and 96.1% percent for BERT. This improvement stems from the high cost of ciphertext operations in larger models, making DictPFL's optimizations more impactful.

*Table 6.* Comparison with baselines on TinyLlama and BERT.

| | Methods | Acc. (%) ↑ | Comm. ↓ | Time ↓ |
|---|---|---|---|---|
| TinyLlama-MetaMathQA | FedHE-Full | 45.86 | 30.0 TB | 214.2 h |
| | FedHE-FT | 6.92 | 2.4 TB | 17.9 h |
| | FedML-HE | 45.86 | 3.0 TB | 22.6 h |
| | DictPFL (ours) | 45.93 | 0.3 TB | 1.3 h |
| BERT-AgNews | FedHE-Full | 91.38 | 137.2 GB | 342.6 m |
| | FedHE-FT | 90.05 | 17.5 GB | 47.9 m |
| | FedML-HE | 91.38 | 13.7 GB | 32.8 m |
| | DictPFL (ours) | 91.24 | 4.8 GB | 13.4 m |

