# OpenReview forum: "DictPFL: Efficient and Private Federated Learning on Encrypted Gradients"
_ICML.cc/2025/Conference — Submitted to ICML 2025_

### Official Review · Reviewer_C4sK · 2025-03-09

**Overall Recommendation:** 2

**Summary:**

This paper studies the problem of using homomorphic encryption in federated learning. The idea is to use Lookup-based Convolutional Neural Network (LCNN), and only encrypt a small fraction of model weights. Furthermore, positions with small scale of gradients are pruned from uploading. Experiments are conducted to show the proposed method can significantly reduce communication and training time.

**Claims And Evidence:**

The paper claims that the privacy is not comprised, which I am not sure if this is right. Due to the usage of pruning, the server can at least know the distribution of large entries in the gradients. One may argue that this does not leak meaningful information, but we still need to make the privacy claim more clear: under the semi-honest setting, which information is protected?

**Essential References Not Discussed:**

N/A.

**Experimental Designs Or Analyses:**

The experiment designs look good to me.

**Methods And Evaluation Criteria:**

I am not sure if the evaluation is fair. The proposed method needs a pretrained weight, and it is not clear how the pretrained weight affect the utility.

**Other Comments Or Suggestions:**

- Figure 2 only have results for existing works. How do your algorithm fit into the figure?
- I am not sure if the use of LCNN restricts the utility, as essentially we are working with a smaller model. Also, since LCNN is not a popular model structure, will this affect the applicability of the proposed method?

**Other Strengths And Weaknesses:**

N/A.

**Questions For Authors:**

- You mentioned secure aggregation [1]. How do you compare your work to secure aggregation? Is the goal to keep the server from knowing aggregated weights? If so, does the following simple alternation of secure aggregation work? The clients first use some crypto-safe protocol to share a random number/matrix $W_{random}$, and perform secure aggregation on $W_{random} + W$, by uploading $W_i + W_{random}/n$. In this way we may not need to perform costly homomorphic encryption.


[1] Bonawitz, Keith, et al. "Practical secure aggregation for privacy-preserving machine learning." proceedings of the 2017 ACM SIGSAC Conference on Computer and Communications Security. 2017.

**Relation To Broader Scientific Literature:**

I think practical HE-based federated learning is an important topic, and this paper makes good progress on it.

**Theoretical Claims:**

This paper does not contain theoretical claims.

---

> ### Author Rebuttal · Authors · 2025-04-01
>
> We thank Reviewer C4sK for providing constructive comments.
>
> **Q1. Why is the gradients privacy not compromised? Pruning may let the server know the distribution of large entries in the gradients. Please clarify what information is protected under the semi-honest setting.**
>
> The gradient's privacy in our DictPFL is not compromised since all model gradients shared with the server are already encrypted, and unencrypted gradients are kept local without sharing. The pruning does not impact privacy since it is performed before encryption. After encryption, the ciphertext of unpruned gradients will not reveal the values or distributions to the server. This ciphertext privacy is guaranteed by FHE. Thus, our methods share the same privacy protection as the baseline with fully encrypted gradients (privacy of gradients and model weights of clients are protected), but are more secure than the prior work FedML-HE that still shares partial unencrypted gradients with the server.
>
> **Q2: It is not sure if the evaluation is fair. The proposed method needs a pretrained weight, and it is not clear how the pretrained weight affect the utility.**
>
> Our evaluations are fairly compared. All methods (DictPFL and our baselines) are conducted on the same pre-trained weights. In the current ML settings, more and more pre-trained models are used, and it is practical to use them as initial models especially for privacy-sensitive fields like healthcare where data is scarce. Starting from a pre-trained model and fine-tuning it on privacy-sensitive data is very promising and practical. In addition, our proposed methods, like Decompose-for-Partial-Encrypt (DePE), work well for cases where no pre-trained model is used. Instead, one could use randomly initialized model parameters, just like LORA work (which decomposes W=A*B) [1] demonstrates strong performance when A is initialized randomly and fixed, and only B is trained. We performed experiments to validate that, to achieve the same level of accuracy (80% for GTSRB dataset with ViT-16), using a randomly initialized dictionary needs 11.81 minutes of training time while the baseline requires 295 minutes of training time. We will clarify this point in the next version of our manuscript.
>
> **Q3. Figure 2 only has results for existing works. How does your algorithm fit into the figure?**
>
> Thanks for asking this question. We used Figure 2 as a motivating example to analyze the execution communication and computation latency breakdown. In Figure 9 of the results section, we put our method and all prior work into one figure for comparison and show that our work significantly reduces the communication and encrypted operations. We will link Figure 2 and Figure 9 in a more explicit way in the next manuscript.
>
> **Q4.  If the use of LCNN restricts the utility, as essentially we are working with a smaller model. Also, since LCNN is not a popular model structure, will this affect the applicability of the proposed method?**
>
> LCNN works more like an advanced weight decomposition technique (not a new architecture), thus when the decomposition is full-rank, it does not reduce the model size or lose representation abilities as shown in equation 1 in the paper.
>
> As an option, users could tune the rank size to control the tradeoff between utility and efficiency. LCNN is not a new model structure or architecture, and it can be applied to any linear projections, convolutions, and Transformers[2][3]. Our experiments have shown that DictPFL works well on models of different scales (from LeNet to Llama2-7b).
>
> **Q5. How to compare your work to secure aggregation? Is goal to keep the server from knowing aggregated weights? If so, does the following simple alternation of secure aggregation work? The clients first use some crypto-safe protocol to share a random number/matrix $W_{random}$, and perform secure aggregation on $W_{random}+W$, by uploading $W_{random}/n+W_i$. In this way we may not need to perform costly homomorphic encryption.**
>
> Yes, one advantage of HE-based FL over Secure Aggregation is to keep the server from knowing aggregated weights (HE will ensure the aggregated weights are still encrypted to server).
>
> Also, the reviewer’s proposed simple alternation of secure aggregation is insightful but introduces additional vulnerabilities. For instance, the server can compute differences between different rounds’ masked weights, i.e., $(W_2+W_{random}/n)-(W_1+W_{random}/n)=W_2-W_1$, to reveal the model updates $\Delta W$ and perform gradient inversion attacks. Additionally, client dropouts disrupt mask cancellation, as missing clients’ shares prevent proper removal of $W_{random}$, corrupting the aggregated result. While promising, this approach requires further innovations to strictly protect weight. This would be an interesting future work.
>
> [1] Improving loRA in privacy-preserving federated learning
>
> [2] Dictformer: Tiny transformer with shared dictionary
>
> [3] Lite-mdetr: A lightweight multi-modal detector

---

> > ### Comment · Reviewer_C4sK · 2025-04-07
> >
> > I would like to thank the authors for their response. Some of my questions have been addressed; however, I remain unconvinced about the necessity of using homomorphic encryption (HE) in the context of federated learning. If the primary goal is merely to prevent the server from accessing the aggregated model weights, then HE may appear excessive. I would appreciate further insights into why HE is essential in this setting.
> >
> > Additionally, I have concerns regarding the reliance on a pre-trained model, as discussed in [1].
> >
> > [1] Tramèr, Florian, Gautam Kamath, and Nicholas Carlini. "Position: Considerations for differentially private learning with large-scale public pretraining." arXiv preprint arXiv:2212.06470 (2022).

---

> > > ### Author Response · Authors · 2025-04-07
> > >
> > > We thank Reviewer C4sK for the valuable questions and comments.
> > >
> > > **Question 1: Necessity of using homomorphic encryption (HE) in federated learning (FL). HE might be excessive if the primary goal is simply preventing the server from accessing aggregated model weights.**
> > >
> > > We clarify that using HE to secure federated learning is not originally proposed by us but has been widely explored and shown to be practical and beneficial in previous literature, including our baselines such as [a] (Zhang et al., 2020) accepted in USENIX ATC, [b] (Roth et al. 2022)  from Nvidia; [c] (Jin et al. 2023) from FedML Inc.. Here we illustrate several important reasons: HE provides comprehensive end-to-end protection—covering model weights/gradients transmission, computation (aggregation), and server storage. This protection addresses multiple security threats, including adversaries in network communications, multi-tenant vulnerabilities during computation on servers, and insider attacks on stored data. HE safeguards not only the confidentiality and intellectual property of model weights and gradients but also protects training data against inversion attacks. Additionally, the overhead of HE in federated learning is often **LESS** significant than commonly assumed. Typically, secure aggregation in FL primarily involves fast HE additions rather than expensive non-linear operations or multiplications. Prior work FedML-HE has demonstrated HE-based FL overheads below 10× compared to plaintext FL (Please see Figure 2). In contrast, our approach further reduces latency overhead to less than 2× (Please see Figure 2 and Figure 9). Hence, we argue that incorporating HE into federated learning is not excessive but rather highly practical and promising.
> > >
> > > **Question 2: Reliance on a pre-trained model, as discussed in [1] of the position paper.**
> > >
> > > The referenced position paper [1] primarily discusses potential privacy risks associated with pre-trained models derived from public datasets, noting situations where publicly available data might inadvertently contain private information. However, it does not argue against the use of pre-trained models entirely. Indeed, initializing models with pre-trained weights for further training offers considerable benefits, most notably substantial reductions in training time compared to training models from scratch with randomly initialized weights. Moreover, our proposed methods, such as Decompose-for-Partial-Encrypt (DePE), do not inherently require pre-trained weights and effectively accommodate various initialization strategies, including random initialization. For instance, we evaluated baselines like FedHE-Full, FedML-HE, and our DictPFL using randomly initialized weights, obtaining training times of 294.6 mins, 56.7 mins, and 11.8 mins, respectively. Although these durations are longer compared to training initialized with pre-trained weights (187.0 mins for FedHE-Full, 29.7 mins for FedML-HE, and 3.1 mins for our DictPFL, as depicted in Figure 9), the results demonstrate our methods' consistent advantage over baselines regardless of initialization. Given the widespread adoption of pre-trained models in current research, relying solely on random initialization might create artificially weak comparisons. Nevertheless, we acknowledge the privacy considerations highlighted in [1] and will incorporate a thorough discussion in the revised manuscript, clearly outlining both the advantages of pre-trained models in reducing training time and the related privacy concerns.

---

### Official Review · Reviewer_Y4dC · 2025-03-12

**Overall Recommendation:** 3

**Summary:**

The paper introduces DictPFL, a novel framework for privacy-preserving federated learning that fully encrypts shared gradients while maintaining efficiency.

**Claims And Evidence:**

Yes

**Essential References Not Discussed:**

No

**Experimental Designs Or Analyses:**

1. Privacy metrics (e.g., LPIPS-based recovery scores) are only applied to image datasets (CIFAR-10, GTSRB, Diabetic Retinopathy). Text tasks (AG’s News, MetaMathQA) lack privacy evaluation, despite gradient inversion attacks potentially leaking sensitive textual data.

2. Results for Llama2 (7B) in Figure 8b raise practicality concerns. Training such models with HE is computationally prohibitive, yet execution details (e.g., rounds, approximations) are omitted.

3. Baselines focus on HE-based methods (FedHE-Full, FedHE-Top2, FedML-HE) but omit non-HE approaches (e.g., sparsity or adaptive pruning techniques). This limits the scope of efficiency comparisons.

4. While ablation studies explore key hyperparameters (e.g., \( r, \tau, s\% \)), HE-specific parameters (e.g., CKKS polynomial degree, scaling factors) are relegated to an inaccessible appendix. Improper settings could compromise security or efficiency.

5. Accuracy drops in Table 3 (e.g., 82.67% → 81.82% as clients increase from 3 to 20) are presented without statistical testing. Small standard deviations (±0.4) suggest robustness, but significance tests (e.g., t-tests) are absent.

6. Key details like total training rounds (main experiments) and Dirichlet \( \alpha \) values (default settings) are unspecified. For example, Table 4 tests \( \alpha \in [0.3, 0.9] \), but the main experiments’ \( \alpha \) is unclear.

7. Data heterogeneity is simulated via Dirichlet sampling, which may not capture real-world non-IID distributions (e.g., user-specific patterns in medical imaging).

**Methods And Evaluation Criteria:**

Yes

**Other Comments Or Suggestions:**

No

**Other Strengths And Weaknesses:**

The security analysis is incomplete, particularly regarding the potential leakage of the globally shared dictionary \( D \) and reactivation patterns in pruning, which could compromise privacy. The technical soundness of the pruning strategy is questionable, as it may be misaligned in non-IID settings due to reliance on historical gradients. Experimental rigor is also a concern, as comparisons with non-HE methods and large-scale evaluations, particularly for text generation tasks, are lacking. The paper does not address key limitations such as the static nature of the dictionary, the impact of pruning on convergence, and the need for clearer algorithmic details and reproducibility.

**Questions For Authors:**

1. Privacy metrics (e.g., LPIPS-based recovery scores) are only applied to image datasets (CIFAR-10, GTSRB, Diabetic Retinopathy). Text tasks (AG’s News, MetaMathQA) lack privacy evaluation, despite gradient inversion attacks potentially leaking sensitive textual data.

2. Results for Llama2 (7B) in Figure 8b raise practicality concerns. Training such models with HE is computationally prohibitive, yet execution details (e.g., rounds, approximations) are omitted.

3. Baselines focus on HE-based methods (FedHE-Full, FedHE-Top2, FedML-HE) but omit non-HE approaches (e.g., sparsity or adaptive pruning techniques). This limits the scope of efficiency comparisons.

4. While ablation studies explore key hyperparameters (e.g., \( r, \tau, s\% \)), HE-specific parameters (e.g., CKKS polynomial degree, scaling factors) are relegated to an inaccessible appendix. Improper settings could compromise security or efficiency.

5. Accuracy drops in Table 3 (e.g., 82.67% → 81.82% as clients increase from 3 to 20) are presented without statistical testing. Small standard deviations (±0.4) suggest robustness, but significance tests (e.g., t-tests) are absent.

6. Key details like total training rounds (main experiments) and Dirichlet \( \alpha \) values (default settings) are unspecified. For example, Table 4 tests \( \alpha \in [0.3, 0.9] \), but the main experiments’ \( \alpha \) is unclear.

7. Data heterogeneity is simulated via Dirichlet sampling, which may not capture real-world non-IID distributions (e.g., user-specific patterns in medical imaging).

8. Claiming "full encryption" is misleading—only lookup tables are encrypted, while the dictionary is shared in plaintext. If \( D \) contains sensitive information (e.g., in medical models), privacy breaches may persist.

9. DictPFL’s efficiency gains depend on pretrained models. Clients without access to such models (e.g., small institutions) cannot participate fairly, exacerbating resource disparities in FL.

**Relation To Broader Scientific Literature:**

The key contributions of the paper are directly related to existing challenges in Federated Learning (FL) and privacy-preserving techniques, particularly regarding the trade-off between privacy and efficiency. Previous works, such as FedML-HE, utilize Homomorphic Encryption (HE) for secure aggregation but suffer from high communication and computational overheads. The paper’s proposed DictPFL framework builds upon these ideas by enhancing HE’s efficiency without sacrificing privacy. It advances the state of the art by introducing two novel modules—DePE and PrME—that reduce the number of encrypted gradients, a key limitation in prior methods.

**Theoretical Claims:**

The paper presents theoretical claims regarding the relationship between the dictionary D and lookup table T in representing the weight matrix W, the reduction in communication overhead, and the privacy preservation of its fully encrypted method. These claims are supported by algorithmic descriptions, equations (such as Equations 1-5), and empirical results but lack formal proofs with lemmas and theorems. While the weight matrix construction and SVD factorization are described mathematically, there are no formal proofs of their optimality or convergence. Similarly, the Temporal Inactivity Pruning and Holistic Reactivation Correction mechanisms include equations for pruning masks and reactivation probabilities but do not provide formal theoretical validation.

---

> ### Author Rebuttal · Authors · 2025-04-01
>
> We thank Reviewer Y4dC for the thorough reading of our manuscript and for providing constructive comments.
>
> **Q1. Misleading "full encryption": Dictionary D and pruning patterns may leak privacy.**
>
> Our "full encryption" refers explicitly to encrypting all information that clients share with the server. The dictionary D is static, untrainable, and globally shared only among clients—it is never shared with or accessible by the server. Reactivation patterns from pruning are based solely on the client's local gradient history, which is also never revealed to the server. The server receives only encrypted pruned gradients, and thus cannot infer pruning or reactivation patterns, ensuring no privacy leakage.
>
> **Q2. Privacy metrics are only applied to image datasets?**
>
> We used Figure 8(a) and privacy metrics (e.g., LPIPS-based recovery scores) as an example to illustrate that FedML-HE leaks sensitive information due to partially plaintext gradients vulnerable to inversion attacks. In contrast, DictPFL encrypts all gradients shared with the server, preventing such privacy leakage for any data type, including images and text.
>
> **Q3. Results for Llama2 (7B) in Figure 8b: execution details (e.g., rounds).**
>
> For the Llama2 (7B) in Figure 8b, we provided the training time for different models, where the training time is equal to the training round number multiplied by the training round time. When we compared the different methods, we calculated the training time required to achieve the same level of performance (60% accuracy on the MetaMathQA task). We will clarify these details in the new manuscript.
>
> **Q4.  Efficiency comparison with non-HE sparsity/pruning methods.**
>
> We compared DictPFL's efficiency with plaintext FL in Figures 2 and 9, showing roughly a 3× overhead for privacy guarantees. Non-HE sparsity or adaptive pruning methods excel in plaintext efficiency but do not provide privacy protections, limiting direct comparison. If reviewers suggest specific methods, we can include additional comparisons.
>
> **Q5. HE parameters: Current compilers ensure the security level for the given HE parameters.** Our HE parameters are secure and are detailed in Appendix A.
>
> **Q6. Table 3 statistical testing.**
>
> We performed Welch’s t-test between the 3-client and 20-client settings in Table 3. The results confirm a statistically significant difference $t(7.0) = 2.50, p = 0.040$.
>
> **Q7. Table 4 tests \alpha in [0.3, 0.9], but the main experiments’ \alpha is unclear.**
>
> The main experiments in Figure 7 were conducted under IID setting (α = ∞). These experiments ran for a maximum of 50 training rounds.
>
> **Q8: Formal proofs with lemmas and theorems.**
>
> Our DictPFL framework includes two HE-aware modules (DePE, PrME) that generalize decomposition and pruning techniques, previously grounded theoretically in plaintext FL. Our primary innovation adapts these established methods to HE constraints rather than developing new theoretical foundations. The practical benefit of our approach lies in significantly reducing ciphertext volume, thus mitigating the high training cost in HE-based FL. We will clarify explicit links to prior theoretical work in the revised manuscript.
>
> **Q9. Pretrained models**: Please refer to C4sK Q2.
>
> **Q10. Study on Client numbers**: Please refer to 4p9J Q2.
>
> **Q11. Text generation tasks.**
>
> In Appendix B.2, we evaluated DictPFL on text generation by fine-tuning the Tiny Llama model on the MetaMathQA mathematical reasoning task. DictPFL reduces training time by 94.2% compared to the previous state-of-the-art, FedML-HE.
>
> **Q12. The pruning strategy convergence (may be misaligned in non-IID settings due to reliance on historical gradients).**
>
> DictPFL addresses pruning misalignment in non-IID settings using Holistic Reactivation Correction (HRC), which reactivates pruned parameters by incorporating client-specific accumulated gradients. This preserves essential local information despite divergence from global patterns. Table 4 shows that HRC effectively maintains stable convergence across varying non-IID settings.
>
> **Q13. Static nature of the dictionary, the impact of pruning on convergence.**
>
> Dictionary decomposition leverages the insight that correlated model weights can be compactly represented as linear combinations of key vectors (dictionary). By using a static dictionary and training only the combination coefficients (lookup tables), we drastically reduce HE-related overhead. Our experiments confirm that this approach achieves higher accuracy and lower training costs compared to previous HE-based methods.
>
> **Q14.  Dirichlet sampling may not capture real-world non-IID distributions.**
>
> Dirichlet is widely used in FL to simulate real-world non-IID. It provides reproducible conditions reflecting practical scenarios like medical imaging [1].
>
> [1] Improving performance of federated learning based medical image analysis in non-iid settings using image augmentation.

---

### Official Review · Reviewer_jD6X · 2025-03-12

**Overall Recommendation:** 3

**Summary:**

The paper proposes DictPFL, a novel framework for federated learning that addresses the trade-off between privacy and efficiency in homomorphic encryption (HE)-based FL. By decomposing model weights into a fixed dictionary and a trainable lookup table (DePE) and further pruning gradients via encryption-aware pruning (PrME), DictPFL significantly reduces communication and computational overhead while ensuring full privacy protection. Experiments demonstrate substantial improvements over fully encrypted and selectively encrypted baselines, achieving up to 748× lower communication overhead and 65× faster training.

**Claims And Evidence:**

yes

**Essential References Not Discussed:**

see below

**Experimental Designs Or Analyses:**

yes

**Methods And Evaluation Criteria:**

yes

**Other Comments Or Suggestions:**

see below

**Other Strengths And Weaknesses:**

Pros:
1. The idea of diction decomposition is novel, and the method seems to be logical and effective.
2. The experiment part is well written, and the way data is shown provides insightful information, which demonstrates the advantage of DictPFL on speed, accuracy and privacy-protection.

Cons:
1. From Section 4.1, DePE is established on the assumption that a public model is shared at the initlal of FL, which may not always hold.
2. TIP and HRC are empirical and lacks novelty.
3. Since the experiments part consider web simulation, it is better to attach codes for more convinient reproduction

**Questions For Authors:**

1. Is it necessary to encrypt indices? What attack will be brought by transmitting encrypted content with plaintext indices?
2. In DePE, is SVD decomposition the only method to reduce matrix rank? Why is T be set as V in SVD decomposition?

**Relation To Broader Scientific Literature:**

see below

**Theoretical Claims:**

yes

---

> ### Author Rebuttal · Authors · 2025-04-01
>
> We thank Reviewer jD6X for the thorough reading of our manuscript and for providing constructive comments.
>
> **Q1. From Section 4.1, DePE is established on the assumption that a public model is shared at the initial of FL, which may not always hold.**
>
> Please refer to the Reviewer C4sK's Q2.
>
> **Q2. TIP and HRC are empirical and lack novelty.**
>
> The proposed TIP and HRC are aiming to address novel and tricky challenges of pruning in HE-based FL. While numerous pruning methods exist in plaintext FL, their direct application to HE-based FL is fundamentally infeasible due to unique cryptographic constraints.
>
> **Key Challenges in HE-Based Pruning**
>
> 1. Server-Side Pruning Limitations: Existing server-side methods [4] require access to plaintext gradients for pruning, which is impossible in HE-based FL, where the server only accesses ciphertexts.
>
> 2. Client-Side Pruning Misalignment: Client-side methods [5] allow clients to prune locally but will lead different clients to prune different positions. And because HE’s SIMD packing mechanism, which encrypts multiple plaintext gradients into a single ciphertext, demands strict alignment of pruned positions. Mismatched indices render aggregated ciphertexts unusable.
>
> In order to address these issues to enable privacy-preserving, HE-compatible pruning, we propose TIP and HRC.
>
> **Novelty of TIP and HRC**
>
> 1. Temporal Inactivity Pruning (TIP) solves the alignment challenge by leveraging global gradient history, which is identical across all clients, to derive a shared pruning mask. This ensures clients prune identical positions, maintaining HE-compatible aggregation.
>
> 2. Holistic Reactivation Correction (HRC) addresses a critical limitation of irreversible pruning: permanently excluding parameters risks losing valuable gradients. HRC dynamically reintroduces pruned parameters based on global importance, preserving model utility without compromising efficiency.
>
> These mechanisms are the first to enable privacy-preserving, HE-compatible pruning, bridging a critical gap between plaintext efficiency techniques and encrypted FL’s constraints. Prior work cannot operate under HE’s limitations, which demand novel solutions to align pruning decisions without exposing sensitive data.
>
> **Q3. Since the experiments part considers web simulation, it is better to attach codes for more convenient reproduction.**
>
> In our submission, we have included the code for the main results as supplementary material to facilitate verification of the results. Upon acceptance, we will release a public GitHub repository with detailed documentation, step-by-step execution guides, and pre-configured environments to ensure the replication of all results.
>
> **Q4. Is it necessary to encrypt indices? What attack will be brought by transmitting encrypted content with plaintext indices?**
>
> In the proposed pruning method PrME, no indices are transmitted, plaintext or ciphertext. Pruning decisions are derived from shared global gradient history, ensuring clients prune identical positions. This eliminates the risk of attacks via plaintext indices (e.g., inferring sensitive patterns from pruned parameter locations). Thus, DictPFL’s design inherently avoids vulnerabilities associated with index transmission while maintaining full encryption of sensitive data.
>
> **Q5. In DePE, is SVD decomposition the only method to reduce matrix rank? Why is T set as V in SVD decomposition?**
>
> While other decomposition methods such as PCA and QR with Truncation can reduce matrix rank, SVD provides the better rank-k approximation according to Eckart-Young theorem, to maintain the important information in the pre-trained weights. Moreover, SVD is widely used to extract low-dimensional information-sensitive representation from model weights to compress the model [1][2][3].
>
> The lookup table $T$ is set to $V$ (right singular vectors) to exploit the orthogonality inherent to SVD. The columns of $V$ form a decorrelated basis for the row space of the original weight matrix, where each column represents an independent direction of variation. This orthogonality minimizes redundancy and captures the most significant parameter correlations, enabling efficient gradient compression. The choice of $T = V$ and $D = U\sum$ (left singular vectors scaled by singular values) aligns with established practices in model compression [3].
>
> [1] Asvd: Activation-aware singular value decomposition for compressing large language models.
>
> [2] Dictformer: Tiny transformer with shared dictionary.
>
> [3] Lite-mdetr: A lightweight multi-modal detector.
>
> [4] Fedmef: towards memory-efficient federated dynamic pruning.
>
> [5] Zerofl: Efficient on-device training for federated learning with local sparsity.

---

### Official Review · Reviewer_4p9J · 2025-03-19

**Overall Recommendation:** 4

**Summary:**

This paper proposes a strategy that selectively encrypts only important weights using Dictionary-based Pruning and Holistic Reactivation Correction (HRC) techniques. This approach maintains the strong security of homomorphic encryption while reducing communication costs and improving training speed. Experimental results show that the proposed method achieves significant improvements compared to conventional methods, with communication cost reductions of 402–748 times and training speed improvements of 28–65 times. Additionally, the method maintains high performance even in WAN environments.

**Claims And Evidence:**

Unlike conventional methods that encrypt all model parameters, this paper proposes a strategy that selectively encrypts only important weights to reduce communication and computational costs. To achieve this, the authors employ Dictionary-based Pruning and Holistic Reactivation Correction to enhance computational efficiency while maintaining security. The server collects encrypted model updates from clients and updates the global model while preserving data privacy and training performance. Experimental results show that the proposed method reduces communication costs by 402–748 times and improves training speed by 28–65 times compared to conventional HE-based federated learning methods while maintaining stable performance in WAN environments.

**Essential References Not Discussed:**

Not applicable.

**Experimental Designs Or Analyses:**

The paper empirically validates that DictPFL achieves superior performance compared to existing HE-based FL methods in terms of communication cost reduction and training speed improvement. The study evaluates performance by comparing it to FedHE-Full and FedML-HE, using key metrics such as communication cost, training speed, model accuracy, security robustness, scalability to large models, and the impact of data distribution and pruning ratio. The experimental design effectively demonstrates the feasibility of DictPFL. However, as the number of clients increases, accuracy slightly decreases. Additionally, security evaluation is limited to Gradient Inversion Attacks, and the paper does not assess its resistance to other security threats. The lack of scalability testing in large-scale client environments and additional security evaluations remain limitations. Future research incorporating experiments with hundreds or thousands of clients and broader security threat evaluations would strengthen the empirical validity of this work.

**Methods And Evaluation Criteria:**

This paper proposes the following methodologies to address the high computational and communication costs in conventional HE-based federated learning:
1. Decompose-for-Partial-Encrypt (DePE): Traditional methods encrypt and transmit the entire model’s weights, which results in high computational and communication overhead. To address this issue, DePE decomposes the model weights into a fixed dictionary and a learnable lookup table, encrypting only the lookup table for transmission. This reduces the amount of encrypted data while maintaining security.
2. Prune-for-Minimum-Encrypt (PrME): Existing methods encrypt a fixed percentage of weights without considering their importance, which may degrade model performance. To improve this, PrME utilizes global gradient information from previous training rounds to prune less important weights first and selectively encrypt only the more critical weights. This reduces unnecessary computations while maintaining model performance. However, in environments with highly imbalanced data distributions (Non-IID settings), the weight selection strategy may introduce model bias, potentially disadvantaging certain clients. Additional measures may be required to address this issue.
The paper evaluates the proposed method’s performance through various experiments, comparing communication cost reduction, training speed improvement, and security maintenance against FedHE-Full and FedML-HE methods. It also experimentally confirms that the approach is scalable to large models, including ViT, BERT, and TinyLlama. Additionally, experiments consider factors such as the number of clients, data imbalance, and pruning ratio, analyzing their impact on model performance and communication efficiency.

**Other Comments Or Suggestions:**

The proposed selective encryption strategy effectively reduces communication and computational overhead in federated learning, as experimentally demonstrated. However, the paper lacks an evaluation of performance variations in large-scale client environments, which is necessary to assess real-world applicability. Further analysis is needed to determine whether the communication cost reduction and training speed improvements persist as the number of clients increases. Additionally, security evaluation is limited to Gradient Inversion Attacks, and further investigation of resistance to Membership Inference Attacks and Model Inversion Attacks would enhance the paper’s contribution. Evaluating the effectiveness of lookup table encryption against diverse attack methods would further strengthen its impact.

This paper proposes an effective selective encryption method that significantly reduces the computational and communication costs associated with HE-based federated learning. The experimental results demonstrate substantial performance improvements, particularly in large models, while maintaining security. The method’s scalability and applicability to ViT, BERT, and TinyLlama strengthen its contribution.
However, the lack of scalability testing in large federated learning environments and the limited security evaluation focused only on Gradient Inversion Attacks remain areas that require further investigation. Addressing these limitations would further improve the practicality of this approach.

**Other Strengths And Weaknesses:**

This paper introduces a selective encryption strategy using DePE and PrME to address the high computational and communication costs of HE-based federated learning, significantly improving both security and efficiency. The paper experimentally verifies that this method is applicable to large-scale models such as ViT, BERT, and TinyLlama, demonstrating its scalability. Compared to FedHE-Full, DictPFL achieves 402–748 times lower communication costs and 28–65 times faster training speeds, making it highly practical. The lookup table encryption method also enhances security by reducing the possibility of original data reconstruction.
However, the experiments do not fully evaluate scalability in large federated learning environments. The security assessment is limited to Gradient Inversion Attacks, and additional evaluations of Membership Inference Attacks or Model Inversion Attacks have not been conducted. These areas require further validation to strengthen the study’s findings.

**Questions For Authors:**

1. Do you have plans for additional experiments to evaluate scalability in large-scale client environments?
2. Do you plan to conduct additional security evaluations against other attack methods?

**Relation To Broader Scientific Literature:**

Existing HE-FL methods provide strong security but suffer from high computational costs and communication overhead. This paper introduces DePE and PrME to implement selective encryption, reducing communication costs by 402–748 times and improving training speed by 28–65 times compared to previous methods, thereby enhancing the practicality of HE-FL. Additionally, the use of lookup table encryption improves the security of conventional methods and enhances resistance to data reconstruction attacks, as demonstrated by experimental results.

**Theoretical Claims:**

This paper presents two key theoretical claims:
1. Decompose-for-Partial-Encrypt (DePE): This method decomposes model weights into a fixed dictionary and a lookup table, encrypting only the lookup table to maintain security while reducing communication costs. The paper claims that this approach significantly reduces encrypted data size compared to traditional methods while making it difficult for attackers to recover the original weights.
2. Prune-for-Minimum-Encrypt (PrME): This method prunes less important weights based on global gradient information from previous training rounds, selectively encrypting only the most important weights to reduce computational and communication costs.

---

> ### Author Rebuttal · Authors · 2025-04-01
>
> We thank Reviewer 4p9J for the thorough reading of our manuscript and for providing constructive comments.
>
> **Q1. Security evaluation is limited to Gradient Inversion Attacks, and the paper does not assess its resistance to other security threats. Do you plan to conduct additional security evaluations against other attack methods?**
>
> We focus on gradient inversion attacks because they represent the primary privacy risk unique to federated learning where attackers exploit gradients during training to reconstruct client data. In contrast, model inversion and membership inference attacks target the final trained model during inference, which is a post-training concern not directly related to the FL process itself. DictPFL’s encryption ensures no gradients are exposed during training, directly addressing the FL-specific threat. Other attacks are model-level risks applicable to any trained model, regardless of how it was trained. These are not influenced by FL’s training protocol. This aligns with prior SOTA HE-based FL work FedML-HE, which similarly focuses on gradient inversion. We will clarify this point in the next manuscript.
>
> **Q2. Do you have plans for additional experiments to evaluate scalability in large-scale client environments?**
>
> We appreciate the reviewer’s suggestion regarding scalability evaluation. In response, we conducted new experiments with 50, 100, and 200 clients, demonstrating that DictPFL consistently outperforms baselines in terms of efficiency. We will incorporate these new results in our new revision.
> | Clients  | FedML-HE (insecure)  | DictPFL (secure)  |
> |----------|----------------------|-------------------|
> | 50       | 18.23 min            | 0.75 min          |
> | 100      | 25.66 min            | 1.22 min          |
> | 200      | 47.05 min            | 1.96 min          |
>
> The training time is evaluated to achieve 81% accuracy on GTSRB using ViT-16 models.

---

### Decision · Program_Chairs · 2025-05-01

**Decision:**

Reject

**Comment:**

This work proposed a new efficient method to perform Federated Learning based on encrypted gradients. Existing HE-based FL methods are either fully encrypted (high cost) or selectively encrypted (leaving shared gradient vulnerable). The proposed provides provide both efficiency and better privacy. It encrypts shared gradients while keeping most gradients local without the need for sharing all, while preserving the performance of global gradient aggregation.

Most of the questions raised by the reviewers were addressed in the rebuttal.
The final reviewers recommendations were 4, 3, 3, 2. The remaining concern from C4sK (2, weak reject) were about the soundness of using HE for Federated Learning.

Overall this is a borderline paper and needs to be revised for future publication.